

# Eastern oysters *Crassostrea virginica* settle near inlets in a lagoonal estuary: spatial and temporal distribution of recruitment in Mid-Atlantic Coastal Bays (Maryland, USA)

Madeline A. Farmer[1], Sabrina A. Klick[2], Daniel W. Cullen[1] and Bradley G. Stevens[1]

[1] Department of Natural Sciences, University of Maryland Eastern Shore, Princess Anne, Maryland, United States

[2] Southeast Watershed Research Laboratory, USDA-ARS, Tifton, Georgia, United States of America

Corresponding author
Madeline A. Farmer,
madelinea.farmer6@gmail.com

## ABSTRACT

**Background:** Declines of the Eastern oyster, *Crassostrea virginica*, and its numerous ecological benefits have spurred oyster restoration initiatives. Successful restoration of a self-sustaining oyster population requires evaluating the temporal and spatial patterns of recruitment (settlement and survival) of oyster larvae in the target waterbody. Restoration of the Eastern oyster population in the Maryland Coastal Bays (MCBs), USA, a shallow lagoonal estuary, is of interest to federal, state, and non-governmental, but the location and timing of natural recruitment is not known.

**Methods:** We assessed the spatial and temporal variation in oyster larval recruitment throughout the MCBs using horizontal ceramic tiles and PVC plates. Newly settled oyster larvae (recruits) were monitored biweekly from June to September 2019 and 2020 at 12 sites in the MCBs and a comparison site in Wachapreague, Virginia. Water quality measurements collected included temperature, salinity, dissolved oxygen, pH, and turbidity. The objectives of this study were to determine (1) the most effective substrate and design for monitoring oyster recruitment, (2) the spatial and temporal distribution of oyster larval recruitment in the MCBs, and (3) patterns in oyster larval recruitment that would be applicable to other lagoonal estuaries.

**Results:** (1) Ceramic tiles were more effective than PVC plates for recruiting oyster larvae. (2) Peak settlement began during the period from late June through July, and oyster recruitment was greatest at sites closest to the Ocean City and Chincoteague inlets. (3) Areas near broodstock that have slow flushing rates to retain larvae may provide the best environments for recruitment of oysters to lagoonal estuaries.

**Discussion:** As the first study on oyster larval recruitment in the MCBs, our results provide insight into their spatial and temporal distribution, methods that can serve as a foundation for future recruitment studies in other lagoonal estuaries, and baseline data that can be used to inform stakeholders and evaluate the success of oyster restoration projects in MCBs.

# INTRODUCTION

Coastal lagoonal estuaries account for 13% of the coastline worldwide (*McAvoy & Clancy, 1994*). The Eastern oyster, *Crassostrea virginica* (Gmelin, 1791), is an inhabitant and keystone species of coastal lagoonal estuaries along the Atlantic shoreline of the United States. Important ecosystem services (*Sanjeeva Raj, 2008*) provided by the Eastern oyster are water filtration (*Wilber & Clarke, 2010*), nitrogen cycling (*Jiang et al., 2020*), benthic-pelagic coupling (*Wazniak, Wells & Hall, 2005*), habitat formation (*Harding & Mann, 2001*), carbon and nitrogen sequestration (*Smyth, Geraldi & Piehler, 2013*; *Fodrie et al., 2017*), and shoreline protection (*Piazza, Banks & La Peyre, 2005*). Despite their ecological importance, anthropogenic stressors such as increased shoreline development, habitat destruction, pollution, water quality degradation (*Lotze et al., 2006*; *Worm et al., 2006*), overharvesting (*Kirby, 2004*), and disease (*Harvell et al., 1999*; *Beck et al., 2011*) have contributed to their population decline.

Population declines and habitat losses of the Eastern oyster have motivated federal, state, and non-governmental agencies to pursue restoration efforts to reestablish the species in native waters (*Chesapeake Bay Program, 2000*). Successful oyster restoration projects have been conducted in lagoonal estuaries along the east coast of the United States at various scales and locations. Small scale restoration projects have occurred in the Delaware Inland Bays, which encompass Rehoboth Bay, Indian River Bay, and Little Assawoman Bay. Volunteers in the Delaware Oyster Gardening Program grow oysters for two years on private docks to be utilized for research or restoration purposes (*Reckenbeil & Ozbay, 2014*). In Maryland and Virginia, federal and state agencies are restoring native Eastern oyster populations and habitats in 10 tributaries (by 2025) as part of the 2014 Chesapeake Bay Watershed Agreement. Of the five tributaries targeted in Maryland, 788 acres of oyster reefs have been restored since 2014, with a goal of 1,439 acres by 2025. Restoration of Eastern oyster populations in the Maryland Coastal Bays (MCBs) have also been discussed among federal, state, non-governmental organizations, and academic partners prior to 2013 (B. Stevens, 2018, personal communication). However, the spatial and temporal distribution of wild oyster larvae in the MCBs was not known.

Surveys on historical oyster bars and adult Eastern oyster populations in the MCBs have been conducted as part of the annual shellfish population surveys by the Maryland Department of Natural Resources (MD DNR) through their Shellfish Monitoring and Assessment Program. Remnant populations of wild Eastern oysters exist in intertidal areas of the MCBs. The Eastern oyster populations have declined dramatically from historic levels due to overharvesting and lasting effects from the creation of the Ocean City Inlet during a hurricane in 1933. This inlet introduced changes in salinity and hydrodynamics within the MCBs as well as new diseases, predators, and competitors (*Tarnowski, 2005*; *Jesien et al., 2009*; *Kang et al., 2017*). Shellfish surveys have never found natural oysters on

the former oyster bars of the MCBs since 1993. Instead, oyster shells are deteriorating, becoming fouled, and buried in sediment (*Tarnowski, 2005*). Although no viable natural oysters exist in subtidal areas of the MCBs, small populations have settled on anthropogenic structures in intertidal areas near Ocean City Inlet and southern Chincoteague Bay where subtidal oyster farms exist (*Tarnowski, 2005*; *Jesien et al., 2009*). Since the annual shellfish surveys only provide data on adult oyster populations, there is an absence of data on the locations of settled oyster larvae, thus spatial and temporal distribution of wild oyster larvae remains unknown in the MCBs.

Prior to restoration initiatives, it is crucial to evaluate the spatial and temporal recruitment patterns of wild oyster larvae as well as their growth and survival over multiple years to determine the feasibility, scale, and location of a restoration effort (*Kennedy et al., 2011*; *Soniat et al., 2012*; *Casas, La Peyre & La Peyre, 2015*). Additionally, identifying the locations of wild oyster larvae are important because natural recruitment supplements and aids in the success of restoration efforts (*Schulte & Burke, 2014*). Therefore, we conducted the first study on the recruitment of oyster larvae in the MCBs and assessed the best methods for measuring recruitment to guide future studies in other waterbodies. From June to September in 2019 and 2020, our objectives were to (1) determine the most effective sampler type (ceramic arrays, PVC arrays, and PVC collectors) for recruitment, (2) determine the spatial and temporal distribution of oyster larval recruitment in the MCBs, and (3) identify patterns in oyster larval settlement that would be applicable to other lagoonal estuaries. Successful oyster restoration in the MCBs would help improve poor water quality (*Jesien et al., 2009*), create hard substrate habitat, and provide additional ecosystem services (*Carruthers & Catherine Wazniak, 2005*; *Jesien et al., 2009*).

## MATERIALS AND METHODS

### Study area

This study was conducted throughout the MCBs, located along the Mid-Atlantic coast of the United States between the Delmarva Peninsula (spanning the states of Delaware, Maryland, and Virginia) and the Atlantic Ocean (*Dennison et al., 2016*). The MCBs system is a shallow lagoonal estuary that encompasses a 453 km$^2$ watershed comprised of six bays ranging from north to south: Assawoman Bay, Saint Martin River, Isle of Wight Bay, Sinepuxent Bay, Newport Bay, and Chincoteague Bay (*Wazniak, Wells & Hall, 2005*; *Krantz et al., 2009*). The MCBs is a two-inlet system with Ocean City Inlet in the north and Chincoteague Inlet in the south. It has an average depth of 1.5 m, but approximately 3 m at Ocean City Inlet and 4 m at Chincoteague Inlet (*Dennison et al., 2016*; *Kang et al., 2017*; *Oseji, Fan & Chigbu, 2019*). As a shallow estuary, it is well-mixed and highly productive with little to no salinity or thermal gradients (*Bricker et al., 2009*; *Oseji, Fan & Chigbu, 2019*).

The MCBs have varied flushing rates, the amount of time it takes for water to be replaced by water exchange through the inlets and freshwater inputs, which range from nine days in Isle of Wight Bay to 63 days in Chincoteague Bay (*Pritchard, 1969*; *Thomas et al., 2009*). Another characteristic of the MCBs is uneven circulation with high current velocities near the inlets that decrease with distance from the inlets (*Krantz et al., 2009*).

The only sources of "new" water (inlets and freshwater input) account for approximately 7.5% of the volume in the MCBs daily (*Pritchard, 1960*). Well-circulated areas have better water quality than areas in or close to tributaries. The uneven distribution of well-circulated areas in combination with input from non-point sources can cause nutrient enrichment that leads to poor water quality (*Bricker et al., 2009*; *Dennison et al., 2016*; *Oseji, Fan & Chigbu, 2019*).

## Site selection

Historical water quality data from the National Park Service (NPS; 2016–2018), MD DNR (1999–2019), and MCBs Program (2013–2015) were used to guide the selection of study sites. From the historical water quality data, twelve sites in the MCBs (10 sites in Maryland and two in Virginia) were selected based on several factors including geographic location, proximity to inlets, salinity (18–39 ppt), bottom type, depth, historical water quality data, and expert recommendations (Fig. 1, Table S1). The study sites ($n = 12$) included three currently monitored for water quality by local agencies: DNR XDN4312 (site: St. Martin River), DNR TUV0021 (site: Turville Creek), and NPS ASSA 2 (site: Verrazano Bridge). Sites had a range of bottom sediments from coarse sand to silt (*Mid-Atlantic Ocean Data Portal, 2021*) and were defined as either Pier or Bay Sites depending on whether sampling equipment was attached to a shore-based pier or placed in open water. An additional study site for sampling gear comparison (substrate and design) was established in Wachapreague, VA, at the Virginia Institute of Marine Science (VIMS) Eastern Shore Laboratory (ESL), where VIMS conducts a recruitment study on oyster larvae.

## Sampler types

"Settlement" is defined as an oyster larvae cementing itself to a substrate, thereby becoming sessile (*Connell, 1985*). "Recruitment" refers to settlement in addition to survival for a time frame defined by the investigator (*Bushek, 1988*; *Roegner & Mann, 1995*). In this study, we defined "recruitment" as recently settled oyster larvae or recruits that survived on settlement substrate for up to two weeks as described in *Rimler (2014)*. Recruitment of oyster larvae was monitored using two different types of sampler designs (collectors and arrays) containing either PVC plates (12.70 cm × 13.97 cm) or ceramic tiles (10.16 cm × 10.16 cm).

PVC collectors consisted of a cage made of plastic-coated wire (22.86 cm × 22.86 cm × 53.34 cm) with 1.5 in$^2$ apertures containing PVC plates and built to our specifications by Ketcham Traps (New Bedford, MA, USA). Each collector contained three PVC plates suspended horizontally using bungee cords at 35.6, 40.6, and 45.7 cm above the substrate and was weighted with two bricks placed in the bottom (Fig. 2A). PVC plates were custom cut to be 12.70 cm × 13.97 cm but had an outer border that confined a counting area to 10.16 cm × 12.70 cm. Plates were drilled in four corners and sanded on both sides with 100 grit sandpaper in a cross-hatched pattern to simulate the rugosity of the outside of an oyster shell to enhance settlement (*Beiras & Widdows, 1995*). An outer border of 6.35 mm on two sides and 12.70 mm on two sides was scored to define a counting area of exactly 10.16 cm × 12.70 cm (129 cm$^2$). The border was defined to ease plate removal and reduce

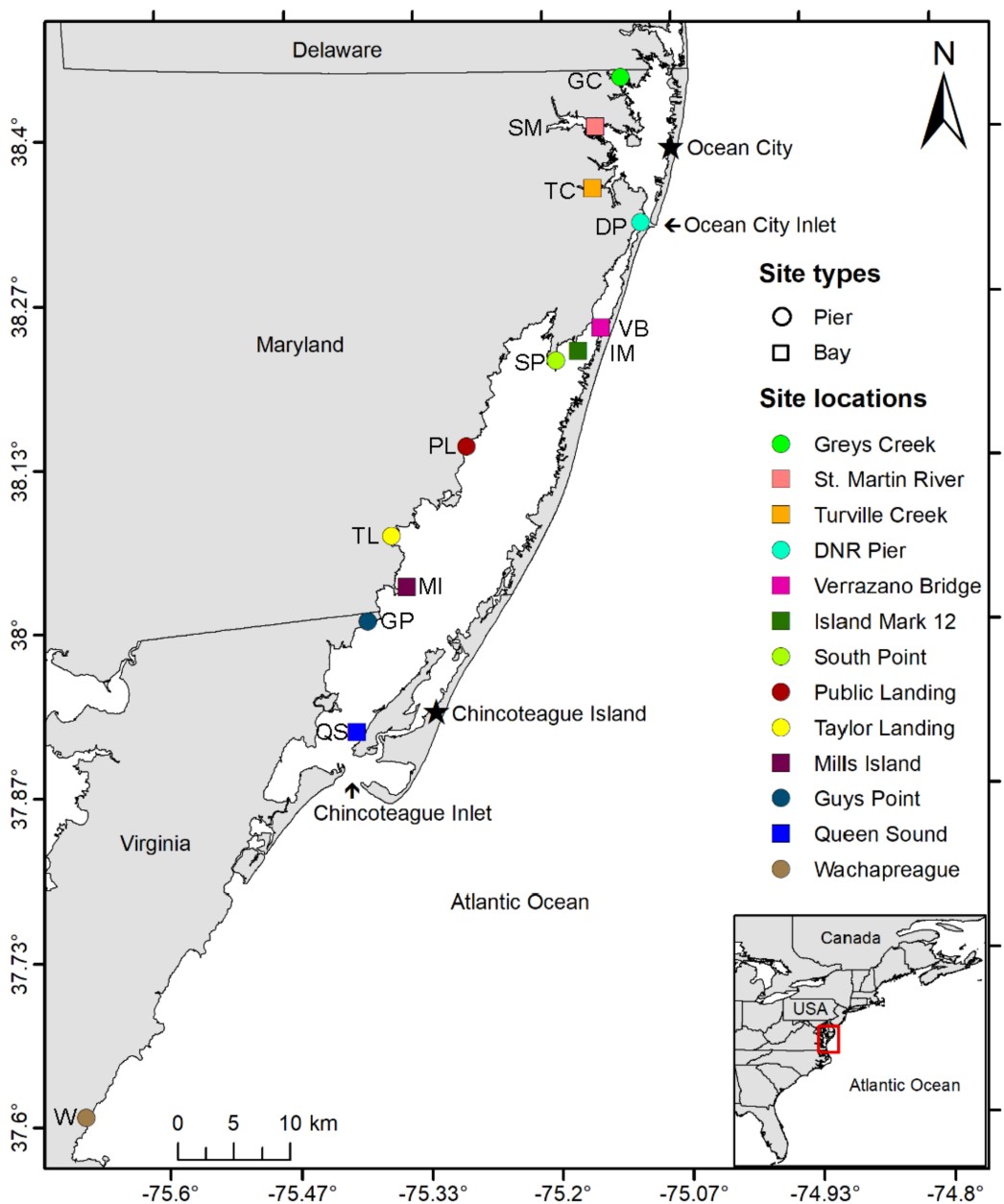

**Figure 1 Study area map.** Map of study area depicting the locations of 10 coastal bay sites in Maryland and three in Virginia where three sampler types were deployed from June to September 2019 and 2020 to assess the distribution of oyster larvae. Circles indicate sites in which sampler types were suspended from a pier while squares indicate sites where sampler types were set on a floating buoy line. Inset shows the location of the study area within the Delmarva Peninsula (USA). Sites include Greys Creek (GC), St. Martin River (SM), Turville Creek (TC), DNR Pier (DP), Verrazano Bridge (VB), Island Mark 12 (IM), South Point (SP), Public Landing (PL), Taylor Landing (TL), Mills Island (MI), Guys Point (GP), Queen Sound (QS), and Wachapreague (W).

the risk of dislodgement of organisms because the plates could not be picked up comfortably using one hand width.

Arrays consisted of a 30.5 cm nylon threaded rod (0.95 cm diameter) on which three center-drilled PVC plates or ceramic tiles (10.2 cm × 10.2 cm × 0.7 cm, 103 cm$^2$) were

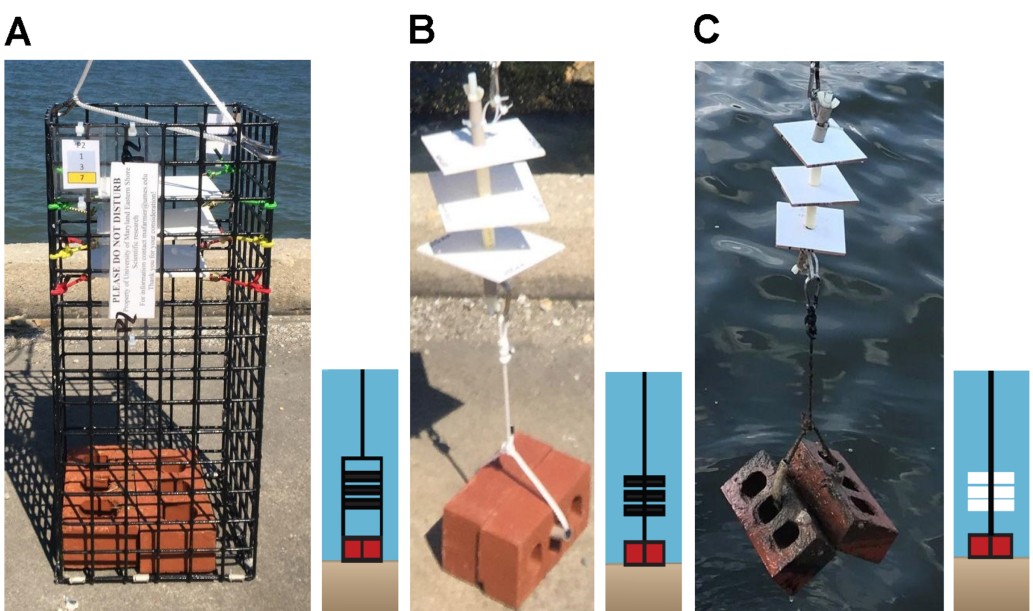

**Figure 2 Sampler types used to monitor recruitment of oyster larvae at 13 sites to assess their distribution.** (A) PVC collector. (B) PVC array. (C) Ceramic array.

positioned and separated by 5 cm sections of 1.25 cm PVC pipe (Figs. 2B, 2C). Arrays did not have a surrounding wire mesh cage. Ceramic tiles were arranged with the unglazed side facing downwards. This design was similar to arrays used by VIMS ESL (*Ross & Snyder, 2020*) but modified with weights (two bricks) below, and a small float (buoy) above, to keep the plates suspended in the water column at a fixed height off the sediment. Ceramic tiles did not have a border, like the PVC plates, because they were smaller and could easily be picked up on the sides using one hand, without risk of dislodging organisms. PVC collectors were deployed in 2019 and 2020 while arrays (both ceramic and PVC) were deployed only in 2020. Because chemical cues have been suggested to induce settlement (*Pawlik, 1986*), approximately 90% of plates and tiles were conditioned in seawater for 8–24 h prior to deployment; the remaining were not due to time constraints.

In 2019, an additional observational study was conducted at VIMS ESL to compare VIMS ceramic arrays and our PVC collector design. Three PVC collectors were suspended next to VIMS arrays made of ceramic tiles. This comparison was made to determine (1) if PVC plates were as effective as ceramic tiles and (2) if a potential lack of recruitment on the PVC plates was due to the collector design, plate substrate, or presence of fewer oysters. Due to the results of this study and low counts in 2019 on PVC plates *vs.* ceramic tiles, we added ceramic array designs to six sites in 2020 as described below. Results from this observational study suggested that ceramic tiles were more effective for monitoring oyster larval recruitment. Therefore, ceramic and PVC arrays employing the VIMS array design were added in 2020 to the sites where recruitment was observed during 2019. Although PVC collectors were less suitable for recruitment than ceramic tiles, they were deployed

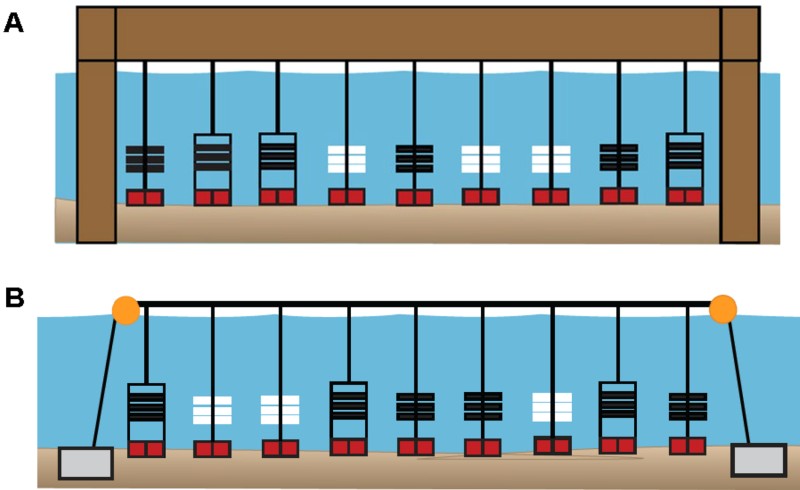

**Figure 3 Diagrams of sampler types deployed at six primary sites in 2020.** (A) Pier site in which ceramic arrays (white), PVC arrays (black), and PVC collectors (black, in rectangles) were suspended from a shore-based pier. (B) Bay site in which sampler types were suspended from a floating buoy line. At both site locations, each sampler type had three replicates that were positioned in random order. In 2019, all 13 sites included three PVC collectors. In 2020, six primary (sites DNR Pier, Island Mark 12, Mills Island, Guys Point, Queen Sound, and Wachapreague) included all three sampler types (ceramic arrays, PVC arrays, and PVC collectors), while the remaining sites included three PVC collectors.

again in 2020 to compare spatial and temporal distribution between years and evaluate potential patterns.

## Field sampling

PVC collectors and arrays were either attached to lines suspended from a shore-based pier (Pier site) or attached to a surface line suspended by buoys between each sampler type (Bay site) (Figs. 3A, 3B). The PVC plates and ceramic tiles were replaced biweekly at the sites for a total of five times per site, or five "swaps" from June–September of 2019 and 2020. In 2019, three PVC collectors (three replicates), each containing three PVC plates ($n = 9$) were deployed at all 13 sites. At all 13 sites in 2019, PVC plates (total $n = 117$) were collected and replaced biweekly ($n = 531$ plates total). During 2020, PVC collectors ($n = 333$ plates), PVC arrays ($n = 225$ plates), and ceramic tiles ($n = 225$ ceramic tiles) could not be deployed at the same five swap dates nor at all 13 sites from 2019 due to COVID-19 and transportation restrictions. Therefore, the "swap dates" typically occurred later and fewer sites were sampled in 2020 than in 2019.

In 2020, PVC collectors were deployed at 11 of the 13 sites, while three PVC arrays and three ceramic arrays were added to six of those sites (designated as "primary sites"). These six sites included four sites where oyster larvae settled in 2019 (Fig. S1), in addition to site Mills Island and Island Mark 12 that were recommended by watermen. Among the six primary sites (DNR Pier, Island Mark 12, Mills Island, Guys Point, Queen Sound, and Wachapreague), plates ($n = 54$) in PVC collectors, plates ($n = 54$) in PVC arrays, and ceramic tiles ($n = 54$) in ceramic arrays were collected and replaced biweekly. Only PVC

collectors with plates were used at the remaining sites including Greys Creek, Verrazano Bridge, South Point, Public Landing, and Taylor Landing. Not all plates and ceramic tiles could be retrieved, however, due to being lost in the field, removed *etc.*

Environmental data were also measured using a Xylem ProDSS Multiparameter Water Quality Meter (Xylem, Yellow Springs, OH, USA) that was positioned above the sediment. Environmental parameters measured included temperature (°C), salinity (ppt), dissolved oxygen (mg/l), pH, and depth (m). Turbidity was measured as secchi disk depth (m). Field experiments were approved by MD DNR under Scientific Collection Permit numbers SCP201964 and SCP202091.

## Laboratory processing

PVC plates and ceramic tiles collected in the field were transported to the laboratory at the University of Maryland Eastern Shore in CD containers to prevent abrasion among plates. Sediment on the plates was gently rinsed and brushed off then a dissecting microscope was used to count oyster larvae. Rugosity of PVC material was consistent on both sides, while ceramic tiles had a smooth glazed top and a rough unglazed bottom. Oyster larvae were counted on the underside of the PVC plates and ceramic tiles to replicate the methods used by VIMS ESL (Wachapreague, VA, USA; *Ross & Snyder, 2020*) and because of the texture differences. Oyster larvae identification was conducted after confirmation by P. G. Ross at VIMS ESL.

## Statistical analysis

All statistical analyses were performed in R version 4.2.2 (*R Core Team, 2022*) and the graphics were generated using the "ggplot2" R package (*Wickham, 2016*). Histogram plots and the Shapiro–Wilk test for normality showed a non-normal distribution for the oyster larval count data. Kruskal–Wallis rank sum tests (*R Core Team, 2022*) followed by Dunn's *post hoc* multiple comparison tests (*Ogle et al., 2022*) were used to determine differences ($\alpha = 0.05$) among sites and sampler types. Due to larval counts producing zero inflated data, the data was subset to counts $\geq 1$ to determine statistical differences among sites and sampler types.

Densities of the larval counts (larval count per $cm^2$ = larval counts from each PVC plate or ceramic tile/area of PVC plate or ceramic tile) were calculated to generate density maps for spatial data visualization. Densities adjusted for plate size differences between ceramic (10.16 cm × 10.16 cm) and PVC plates (10.16 cm × 12.70 cm).

Larval counts and water quality measurements from all sites 2019 ($n = 496$) and 2020 ($n = 830$) were combined into one dataset to examine potential relationships. The larval counts from plates/tiles and sampler types were summed to obtain independent observations of larval counts and water quality measurements from sampling time points within each site ($n = 13$). The correlations between the larval counts and water quality measurements in the combined ($n = 58$) and 2020 ($n = 47$) dataset were evaluated using principal component analysis (PCA). The significance of these correlations was tested using the Kendall Tau-b ($\tau_B$) rank correlation method ($\alpha = 0.05$; *McLeod, 2011*). This correlation method is appropriate when data contains non-normal distributions, tied

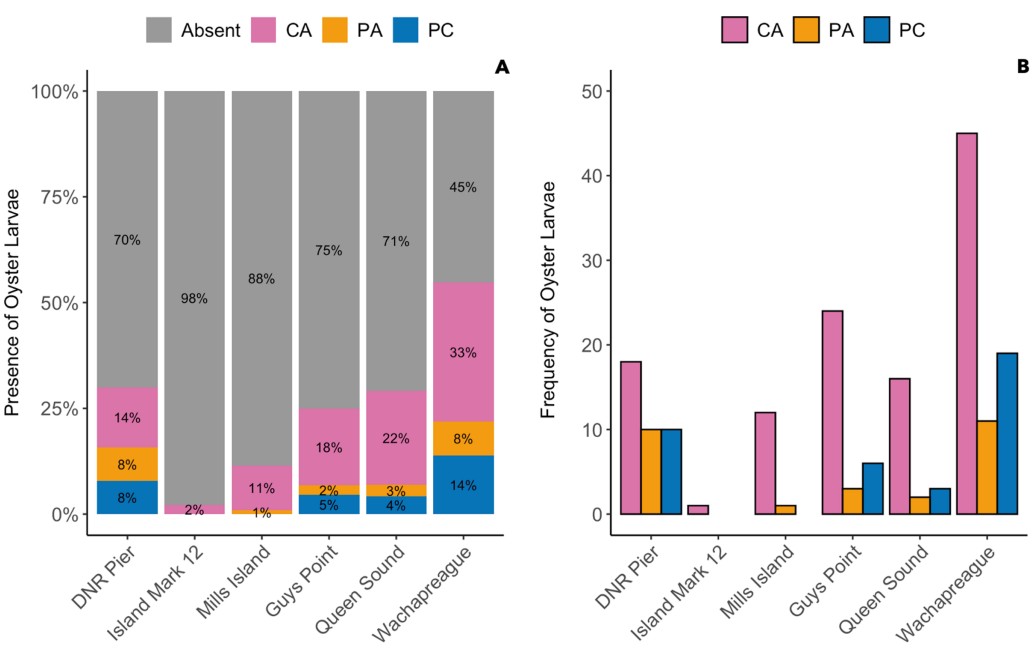

**Figure 4** **The proportion of the presence and absence of oyster larvae and the number of times oyster larvae was present on the sampler types at each site during 2020.** The larval count data was used to calculate the presence and absence of oyster larvae on sampler types from each site. (A) The proportion of oyster larvae that was absent (gray) or present on ceramic arrays (CA; pink), PVC arrays (PA; orange), and PVC collectors (PC; blue). Sites and total number of observations were DNR Pier ($n = 127$), Island Mark 12 ($n = 45$), Mills Island ($n = 113$), Guys Point ($n = 132$), Queen Sound ($n = 72$), and Wachapreague ($n = 137$). (B) The number of times (frequency) that oyster larvae was observed on the sampler types.

ranks, and outliers (*Croux & Dehon, 2010*; *Alfons, Croux & Filzmoser, 2017*; *Akoglu, 2018*). The PCA plot was generated using the "prcomp" function in the FactoMineR package (*Kassambara & Mundt, 2020*), which uses singular value decomposition to examine covariances and correlations between the observations. The factoextra package (*Lê, Josse & Husson, 2008*) was used to evaluate the eigenvalues to determine the highest percentages of variance retained by each principal component.

To determine the influence of sampler type, site location, and sample timing on larval recruitment, the oyster larval counts from 2019 and 2020 were used to generate a generalized linear mixed model (GLMM) with the "glmmTMB" R package (*Brooks et al., 2017*). Models were run with a zero-inflated Poisson (ZIP) regression due to the high percentage of zeros (84.5%) in the dataset. The Akaike information criterion (AIC) values from each model (m$i$) were used to calculate a second-order bias correction estimator (AIC$_C$). A model was chosen based on the AIC$_C$ values and quality checks provided by the "DHARMa" R package (Fig. S2; *Hartig, 2022*).

## RESULTS

### Sampler types

The presence of oyster larvae varied by sampler type and site during 2020 with many zero counts (absence) within sites (Fig. 4A). Ceramic arrays were the most effective sampler

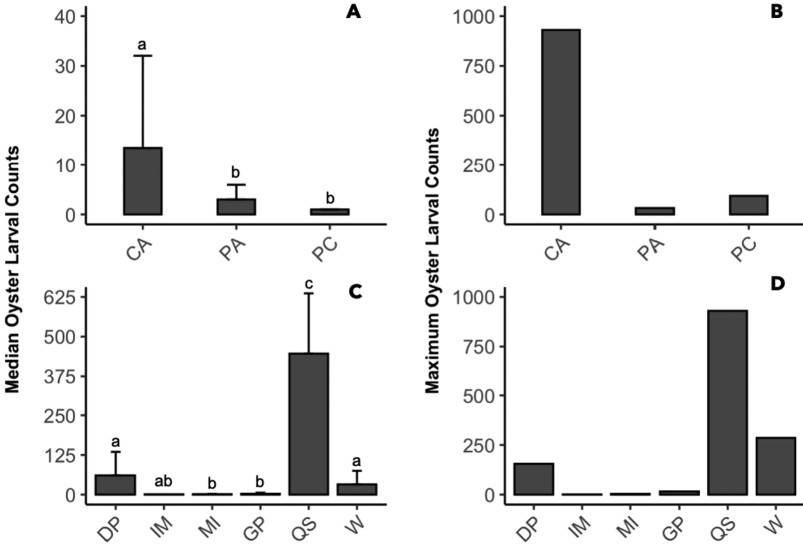

**Figure 5 Comparison of median and maximum larval counts to sampler types and sites during 2020.**
Data was pooled by either sites or sampler types to compare the median and maximum larval counts
(≥1). (A, B) Counts from sampler types of ceramic arrays (CA; $n = 116$), PVC arrays (PA; $n = 27$), and
PVC collectors (PC; $n = 38$). (C, D) Counts from sites of DNR Pier (DP; $n = 18$), Island Mark 12 (IM;
$n = 1$), Mills Island (MI; $n = 12$), Guys Point (GP; $n = 24$), Queen Sound (QS; $n = 16$), and Wachapreague
(W; $n = 45$). Bars above median counts represent median absolute deviation (MAD). The letters above
the deviation bars denote significant differences ($p < 0.05$) between median larval counts.

type for assessing oyster larval recruitment at the six primary sites in the MCBs. Oyster
larvae settled on 2–33% of the ceramic tiles and on 1–14% of the PVC plates (Fig. 4A).
Additionally, the settlement of oyster larvae was observed more frequently on the ceramic
arrays (116 times) than on the PVC arrays and PVC collectors combined (65 times) among
the sites (Fig. 4B; $H(2) = 7.054$, $p = 0.029$). The median larval counts were also significantly
higher on ceramic arrays (13.5 ± 18.53 [MAD]) compared to PVC collectors (1 ± 0
[MAD]) and PVC arrays (3 ± 2.97 [MAD]; Fig. 5A; H (2) = 34.393, $p < 0.0001$).
The ceramic arrays had larval counts up to 930 compared to the PVC arrays and PVC
collectors with counts up to 32 and 93, respectively (Fig. 5B). The oyster larvae on the
ceramic arrays were observed most frequently at DNR Pier (18), Guys Point (24), Queen
Sound (16), and Wachapreague (45; Fig. 4B). The median larval count (447 ± 189.78
[MAD]) was significantly higher at Queen Sound compared to the other five sites, but
maximum larval counts >100 were observed at Queen Sound, DNR Pier, and
Wachapreague (Figs. 5C, 5D; $H(5) = 68.855$, $p < 0.05$).

## Spatial distribution
PVC collectors were deployed during 2019 and 2020 to compare distribution patterns in
oyster recruitment between years. No oyster settlement occurred at sites other than the six
primary sites during 2019 and 2020, except one occurrence at North Verrazano Bridge
during August 2020 (Table 1). Of the six primary sites, oyster settlement occurred in both
years at DNR Pier, Guys Point, Queen Sound, and Wachapreague. Little recruitment

**Table 1 Total oyster larval counts and water quality measurements of sites during 2019 and 2020.** Larval counts were summed by site to calculate totals and water quality measurements of depth, temperature, salinity, pH, dissolved oxygen, and turbidity. Values were averaged (mean ± standard deviation) by site across years and time.

| Site | Site type | $n$ | Total larval counts | Depth (m) | Temperature (°C) | Salinity (ppt) | pH | Dissolved oxygen (mg O L$^{-1}$) | Turbidity (m) |
|---|---|---|---|---|---|---|---|---|---|
| Greys Creek | Pier | 10 | 0 | 0.61 ± 0.09 | 28.80 ± 1.22 | 24.86 ± 2.09 | 7.65 ± 0.41 | 4.19 ± 2.80 | 0.44 ± 0.12 |
| St. Martin River | Bay | 10 | 0 | 0.95 ± 0.13 | 27.79 ± 2.18 | 26.58 ± 0.91 | 7.85 ± 0.33 | 6.54 ± 0.81 | 0.45 ± 0.05 |
| Turville Creek | Bay | 8 | 0 | 0.69 ± 0.25 | 28.40 ± 1.83 | 26.62 ± 1.14 | 7.77 ± 0.43 | 6.96 ± 1.01 | 0.45 ± 0.13 |
| DNR Pier | Pier | 7 | 1,374 | 2.81 ± 0.60 | 22.54 ± 2.04 | 30.40 ± 1.29 | 8.10 ± 0.38 | 6.64 ± 0.76 | 0.95 ± 0.28 |
| Verrazano Bridge | Bay | 4 | 3 | 1.09 ± 0.29 | 27.16 ± 1.04 | 29.07 ± 1.36 | 8.18 ± 0.47 | 7.57 ± 1.28 | 0.61 ± 0.12 |
| Island Mark 12 | Bay | 8 | 1 | 0.87 ± 0.14 | 26.84 ± 1.70 | 28.69 ± 1.82 | 8.14 ± 0.31 | 7.02 ± 0.81 | 0.49 ± 0.11 |
| South Point | Pier | 7 | 0 | 0.91 ± 0.12 | 28.43 ± 1.11 | 28.97 ± 2.00 | 8.08 ± 0.33 | 6.42 ± 1.12 | 0.43 ± 0.20 |
| Public Landing | Pier | 10 | 0 | 0.76 ± 0.12 | 28.72 ± 0.98 | 28.20 ± 2.08 | 7.99 ± 0.31 | 6.54 ± 0.94 | 0.49 ± 0.15 |
| Taylor Landing | Pier | 8 | 0 | 0.78 ± 0.16 | 28.81 ± 1.89 | 29.35 ± 2.21 | 7.94 ± 0.34 | 5.75 ± 1.42 | 0.37 ± 0.11 |
| Mills Island | Bay | 10 | 22 | 0.88 ± 0.15 | 26.82 ± 1.49 | 30.88 ± 1.65 | 7.96 ± 0.36 | 5.93 ± 0.94 | 0.42 ± 0.12 |
| Guys Point | Pier | 10 | 125 | 1.20 ± 0.25 | 29.61 ± 1.01 | 30.93 ± 1.48 | 7.87 ± 0.26 | 6.12 ± 1.99 | 0.59 ± 0.27 |
| Queen Sound | Bay | 4 | 6,912 | 1.25 ± 0.14 | 26.61 ± 1.92 | 30.96 ± 1.25 | 8.01 ± 0.39 | 6.41 ± 0.79 | 0.54 ± 0.09 |
| Wachapreague | Bay | 9 | 3,391 | 1.44 ± 0.59 | 28.26 ± 1.22 | 32.22 ± 1.29 | 7.52 ± 0.12 | 4.00 ± 0.63 | 0.38 ± 0.11 |

occurred at Island Mark 12 and Mills Island (Table 1). Spatial patterns of recruitment on PVC collectors were consistent between 2019 and 2020 because all sites with settlement in 2019 also received settlement in 2020 (Fig. 6A). Total settlement density from all sites on the PVC collectors was greater during 2020 (3.06 larvae per cm$^2$) than 2019 (0.37 larvae per cm$^2$). At DNR Pier, settlement density was five times greater on the PVC collectors during 2020 (0.50 larvae per cm$^2$) than 2019 (0.11 larvae per cm$^2$; Fig. 6A). Additionally, total settlement density from 2019 and 2020 at Wachapreague (2.62 larvae per cm$^2$) was almost 4 times greater than settlement at DNR Pier (0.61 larvae per cm$^2$). Interannual differences were only significant at Wachapreague ($H$ (1) = 4.95, $p$ = 0.0204) but not at DNR Pier or the primary sites. No recruitment occurred at Island Mark 12 and Mills Island for PVC collectors during both years. Recruitment on the PVC collectors was low at Queen Sound in both years and recruitment was slightly less during 2020 (0.03 larvae per cm$^2$) than 2019 (0.07 larvae per cm$^2$; Fig. 6A). Recruitment at Guys Point (0.05 larvae per cm$^2$) remained the same for both years.

The spatial distribution of settlement densities from the ceramic arrays were compared to the settlement densities from the PVC arrays, which were only deployed during 2020 at the six primary sites (Fig. 6B). Overall, total settlement density was greater on ceramic arrays (109 larvae per cm$^2$) than PVC arrays (0.93 larvae per cm$^2$; $H$ (1) = 96.291, $p$ < 0.0001). Within the MCBs, the greatest larval density on ceramic arrays occurred at Queen Sound (66.81 larvae per cm$^2$), Wachapreague (29.31 larvae per cm$^2$), and DNR Pier (11.73 larvae per cm$^2$) in 2020 over the entire field season (Fig. 6B). Among the six primary sites, settlement density on the ceramic arrays was greatest at the sites closest to the inlets (DNR Pier and Queen Sound). Lastly, Island Mark 12 (0.01 larvae per cm$^2$) and Mills

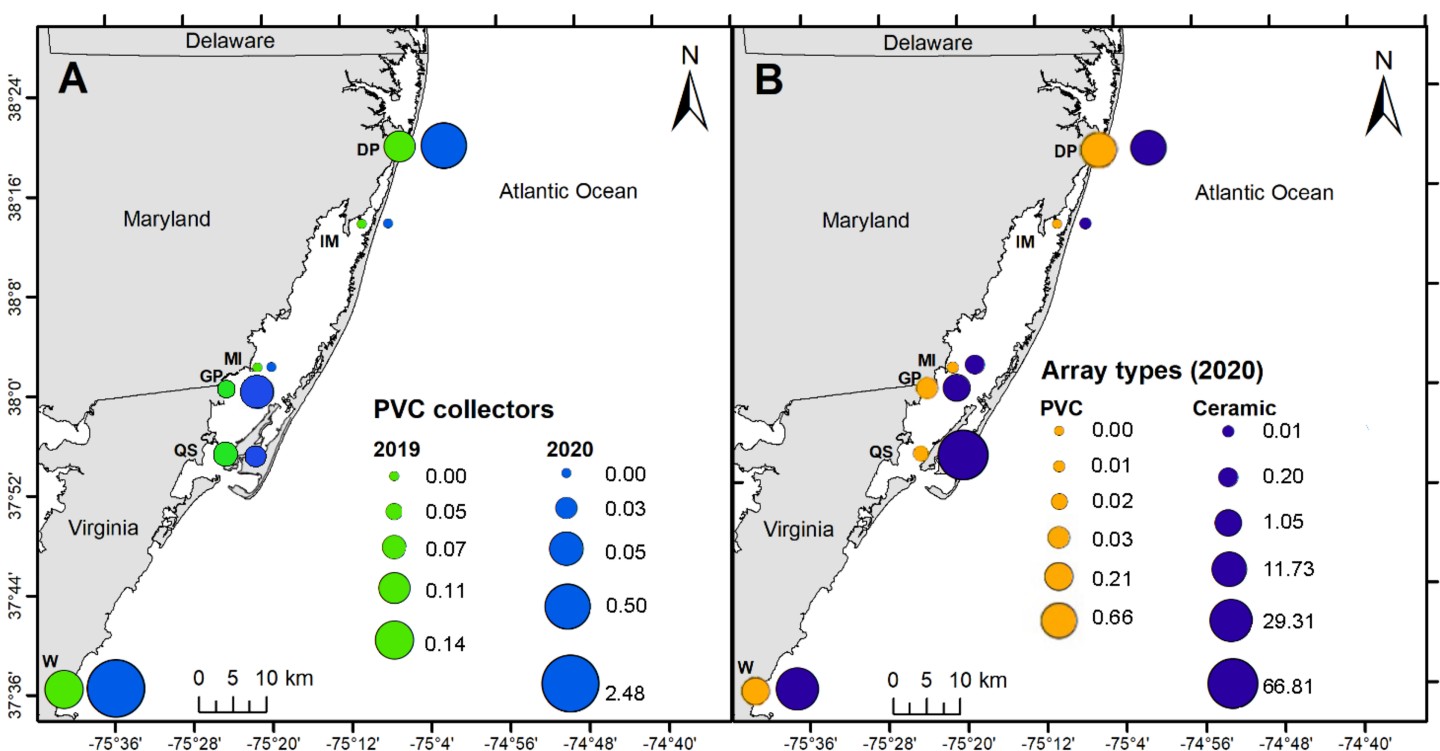

**Figure 6 Map of spatial distribution of oyster larvae that settled on three sampler types at six primary coastal bay sites.** (A) Density of oyster larvae per plate/tile over entire field season on PVC collectors in 2019 and 2020. (B) Density of oyster larvae per plate/tile over entire field season on PVC arrays and ceramic arrays in 2020. The values and circle sizes represent the total density of oyster larvae from the underside of the ceramic tiles or PVC plates. Total density was calculated by dividing the larval counts by the plate/tile area and then summing the density values at each site by year or sampler type. Six primary sites included DNR Pier (DP), Island Mark 12 (IM), Mills Island (MI), Guys Point (GP), Queen Sound (QS), and Wachapreague (W).

Island (0.21 larvae per cm²) received little recruitment when both arrays were summed (Fig. 6B).

## Temporal distribution

At all sites in the MCBs and Wachapreague, settlement in 2019 and 2020 generally began in early to mid-July, but settlement only occurred in late-July of 2019 at Island Mark 12 (Fig. 7) Settlement at the MCB sites continued until late July to mid-August during both years. In 2020, settlement at sites Wachapreague and Queen Sound occurred earlier than the remaining primary sites (Figs. 7D, 7E). Earliest settlement within the MCBs occurred at site Queen Sound, but sampling equipment at that site disappeared after 30 July 2020 due to a storm, which prevented further data collection (Fig. 7D). Settlement began slightly earlier at site Wachapreague than in the MCBs, in late June (2019) and early July (2020), and extended longer, until late August in 2020 (Fig. 7E). At site DNR Pier, near Ocean City Inlet, two settlement peaks were observed in both 2019 and 2020 and occurred within approximately the same week of each year (Fig. 7A).

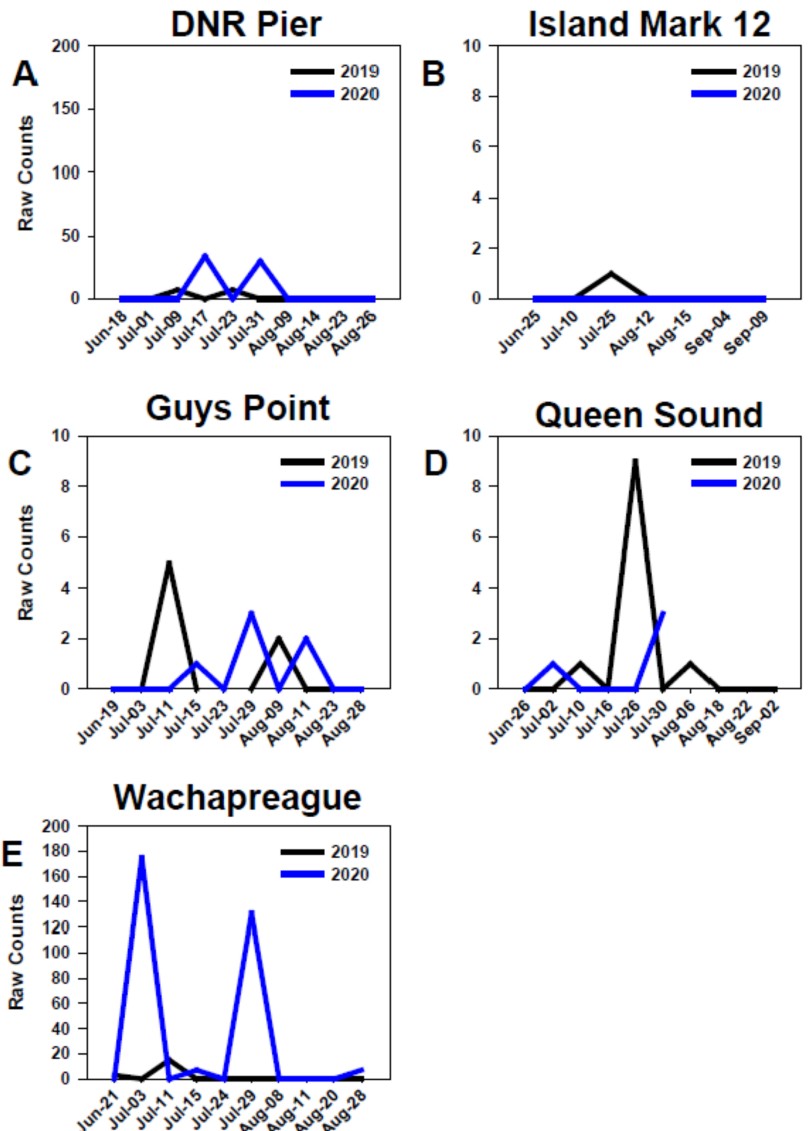

**Figure 7 Temporal distribution of oyster larvae at coastal bay sites.** Comparison of recruitment on PVC collectors at five sites from June to September 2019 and 2020. Figures display raw counts of oyster larvae counted on the underside of PVC plates within PVC collectors. Sites (A) DNR Pier. (B) Island Mark 12. (C) Guys Point. (D) Queen Sound. (E) Wachapreague. Note scales of y-axes differ.

## GLMM model for larval recruitment prediction

Several ZIP GLMMs were tested and compared against a null model to determine the best fit model for predicting larval counts based on sampler type (St), site location (Si), and sample timing represented as swap number (Sw; Table 2). The $m_4$ model was rejected because sampler type, site location, and sample timing as fixed effects increased the AICc by 702.5, reduced the probability ($w_i$) to <0.05, and failed quality check analyses (outlier test and Kolmogorov-Smirnov test for uniformity of the residuals). Models $m_6$ and $m_7$ results had the lowest $AIC_C$ values and uniquely contained the plate/tile levels within the

**Table 2 Comparison of ZIP generalized linear mixed models ($m_0$–$m_7$) corresponding to the different variables tested for oyster larval settlement prediction.**

| Model | Variables | df | logLik | AIC$_C$ | $\Delta_i$ | $w_i$ |
|---|---|---|---|---|---|---|
| $m_0$ | L ~ 1 | 2 | −18,161.9 | 36,327.9 | +30,624.8 | 0 |
| $m_1$ | L ~ Sw + (1\|ST) + (1\|Si) + (1\|N) | 9 | −3,217.5 | 6,453.1 | +750.0 | <0.001 |
| $m_2$ | L ~ Si + (1\|ST) + (1\|Sn) + (1\|N) | 17 | −3,205.7 | 6,445.9 | +742.8 | <0.001 |
| $m_3$ | L ~ Si + Sw + (1\|ST) + (1\|N) | 20 | −3,189.8 | 6,420.2 | +717.1 | <0.001 |
| $m_4$ | L ~ Si + Sw + ST + (1\|N) | 21 | −3,181.5 | 6,405.6 | +702.5 | <0.001 |
| $m_5$ | L ~ Si + Sw + (1\|ST) + (1\|N) + (1\|L) | 21 | −3,014.8 | 6,072.3 | +369.2 | <0.001 |
| $m_6$ | L ~ Si + (1\|Sw) + (1\|ST) + (L\|N) | 22 | −2,842.0 | 5,728.8 | +25.7 | <0.001 |
| $m_7$ | L ~ Si + Sw + (1\|ST) + (L\|N) | 25 | −2,826.1 | 5,703.1 | – | 0.99 |

**Note:**

df, degrees of freedom; Loglik, log-likelihood; AIC$_C$, corrected AIC value; $\Delta_i$, difference between each model and the best selected model; and $w_i$, probability that a given model provided is the best fit for the data. Variables: L, oyster larvae counts; Si, site; Sw, swap number represented sampling time; ST, sampler type; L, plate level on the sampler; N, line number represented the position of the sampler on the pier or buoy line. Model $m_7$ was selected to be the best fit model.

samplers and line number, the location of the sampler on the pier or buoy line, as a nested random effect. We chose model $m_7$, with site (Si) and time (Sw) as fixed effects, as the best fit model because it resulted in the lowest AIC$_C$ value, a probability >0.05, and passed model quality checks (Table 2; Fig. S2). The best fit model, $m_7$, succesfully  predicted the highest larval counts to occur at DNR pier, Queen Sound, and Wachapreague sites and during swaps 2 and 3 which represented the time period between late June and mid-August (Fig. S3).

## Environmental effects

The principal component analysis (PCA) plot showed potential relationships between oyster larval counts and water quality measurements of temperature, salinity, dissolved oxygen (DO), pH, and turbidity. The principal components (PC-1 and PC-2), calculated from the water quality variables and larval counts, explained 59% of the variance observed in the dataset (Fig. 8). Results of the Kendall Tau-b ($\tau_B$) rank correlation tests showed that larval counts had a significant positive correlation with salinity ($\tau_B = 0.49$, $z = 6.545$, $p < 0.0001$) and negative correlations with DO ($\tau_B = -0.17$, $z = -2.237$, $p < 0.05$) and pH ($\tau_B = -0.19$, $z = -2.453$, $p < 0.05$). Average salinity ranged from 24.86–32.22 ppt among sites and was highest (>30 ppt) at DNR pier, Mills Island, Guys Point, Queen Sound, and Wachapreague (Table 1). At these five sites, the average pH and DO were 7.52–8.10 and 4.0–6.64 mg O L$^{-1}$, respectively (Table 1). No correlations were found between the larval counts and temperature or turbidity in the PCA analysis. The average temperature and turbidity among these five sites ranged 22.54–29.61 °C and 0.38–0.95 m, respectively (Table 1).

## DISCUSSION

### Sampler types

Oyster larvae exhibited preferential settlement (*Keough & Downes, 1982*), indicated by significantly more counts on the ceramic arrays in 2020 than any other sampler type.

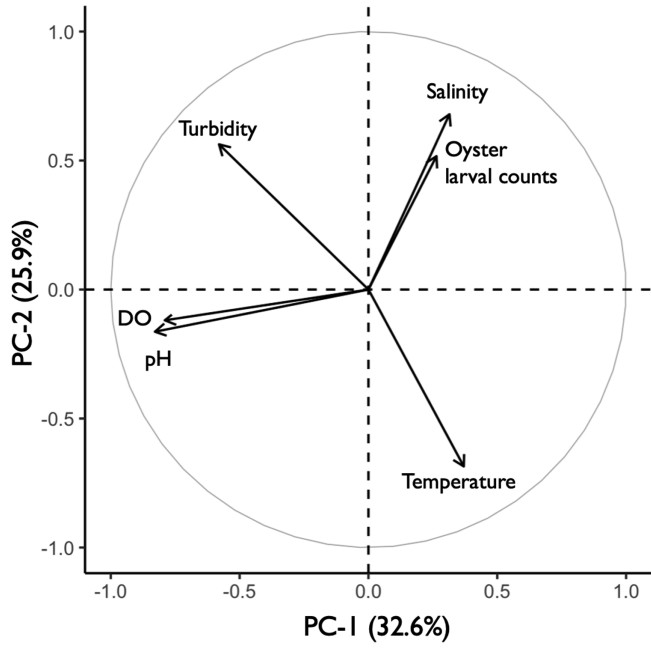

**Figure 8** **Principal component analysis (PCA) of oyster larval counts and water quality parameters.** PCA plot showing the potential correlations between oyster larval counts and water quality measurements of salinity, turbidity, dissolved oxygen (DO), pH, and temperature (°C). The correlations between larval counts and water quality measurements were obtained by combining the 2019 ($n = 58$) and 2020 ($n = 47$) datasets with all 13 sites.

Despite our PVC plates being sanded with 100 grit sandpaper, the ceramic tiles had greater rugosity, making it easier for oyster larvae to attach (*Marques-Silva et al., 2006*).
In addition, ceramic tiles are alkaline (*Reig et al., 2013*) and oyster larvae are more likely to settle when exposed to ammonia, which is alkaline (*Coon, Fitt & Bonar, 1990*). Preferential settlement on ceramic tiles rather than PVC was also evident in the study by *Chuku et al. (2020)*, who compared monthly recruitment of the West African mangrove oyster *Crassostrea tulipa* among five substrates (coconut shell, oyster shell, nylon mesh, PVC slats, and ceramic tile) in four lagoonal estuaries in Ghana. Ceramic tiles had the greatest monthly settlement in three of the four estuaries and PVC slats had the greatest in only one of the estuaries, but overall recruitment was not significantly different between ceramic tiles and PVC slats (*Chuku et al., 2020*). The presence of biofilms on attachment substrates can enhance oyster settlement (*Tamburri, Zimmer-Faust & Tamplin, 1992*; *Zhao, Zhang & Qian, 2003*; *Su et al., 2007*; *Campbell et al., 2011*), but the effect varies with length of conditioning and rugosity of the substrate (*Taylor, Southgate & Rose, 1998*; *Devakie & Ali, 2002*; *Zhao, Zhang & Qian, 2003*; *Su et al., 2007*; *Tamburri et al., 2008*; *Bellou et al., 2020*).

## Spatial distribution

Although settlement and recruitment behaviors can be difficult to measure *in-situ*, our results show a spatial distribution trend of greater recruitment at sites near Ocean City (DNR Pier) and Chincoteague Inlet (Queen Sound), suggesting those are more attractive locations for oyster settlement than sites further away from inlets. Additionally,

broodstock live close to those inlets. Settlement of both Sydney rock oyster *Saccostrea glomerata* and invasive Pacific oysters *Crassostrea gigas* in Port Jackson Estuary, Australia was greater at sites closer to the Pacific Ocean than in the upper channel (*Scanes et al., 2016*). Spatial distribution patterns in the Port Jackson Estuary were similar to those of the MCBs with more observations of oyster larvae closer to the interface between the estuary and ocean.

The MCBs have lower freshwater discharge, varied flushing rates, and high velocity water near Ocean City and Chincoteague inlets. Flushing rates in the individual sub-bays vary greatly, *e.g.*, from 9 days in Isle of Wight Bay to 63 days in Chincoteague Bay (*Pritchard, 1969*; *Lung, 1994*). This implies that oyster larvae are retained longer, thus have a longer period of time to settle in Chincoteague Bay than in Isle of Wight Bay. The longer retainment period in Chincoteague Bay supports the greater settlement observed near Chincoteague Inlet (*Wazniak, 2005*). Retention within a system correlates to recruitment success (*Norcross & Shaw, 1984*).

A hydrodynamic model by *Kang et al. (2017)* demonstrated that the northward flow of water through the MCBs is primarily wind-driven, except when wind speeds are weak (*e.g.*, 3 m/s), at which times it becomes tidally driven. However, tidal cycles drive the circulation patterns near the Ocean City and Chincoteague inlets (*Wells, Hennessee & Hill, 2002*). The MCBs have a distinct seasonal wind pattern of prevailing winds from the southwest in the summer, due to the Bermuda High pressure system, and from the northeast in the winter. Circulation patterns in the MCBs may also be influenced by the shape of the estuary's basin and bathymetry, or depth (*Lee & Valle-Levinson, 2012*) where its shallow basin and wind patterns can alter wave dynamics (*Mao & Xia, 2018*). Strong turbulence from waves can cue oyster larvae to sink, increasing their proximity to suitable substrate in which to attach (*Fuchs et al., 2013*). The strong turbulence at the inlets could be a reason for the observed spatial distribution.

The MCBs are characterized as being "microtidal" since tidal exchange is limited to Ocean City Inlet and Chincoteague Inlet. Although the tidal excursion of the MCBs is unknown, similar lagoonal estuary systems have tidal excursions of 2.7 km in Haulover Canal connecting Mosquito Lagoon and Indian River lagoon (*Smith, 1993*), 2 km both for Little Egg Harbor and Barnegat Bays (*Chant, 2001*), and 1.02 to 8.25 km depending on the site proximity to Fort Pierce Inlet within the Indian River lagoon (*Smith, 1983*). Tidal excursion refers to the distance between low water and high water in which a particle travels. It is a measurement to describe the movement of particles such as larvae and pollutants within a tidal cycle (*Ji, 2008*). A coupled biological-physical transport model by *Kim, Park & Powers (2013)* simulated that larger tidal excursions during a tropic tide caused greater larval dispersion. We hypothesize that the spawning adult oysters are within a 1–8 km range of settlement sites in the MCBs. Perhaps the coupling of flushing rates (slowest in Chincoteague Bay), proximity, and tidal circulation near the inlets may have contributed to greater settlement.

## Temporal distribution

The Eastern oysters in this study region typically spawn from June through October (*Haven & Fritz, 1985*). Sampler types showed little variation in peak timing between 2019 and 2020 (<10 days apart) and all settlement peaks occurred in July during both years. Our results showed that oyster larvae settled between late-June and mid-August, which was expected based on settlement timing reported by previous studies at similar latitudes in the Mid-Atlantic (*Shaw, 1967*; *Kennedy, 1980*; *Haven & Fritz, 1985*; *Capelle et al., 2020*; *Ross & Snyder, 2020*). Although monitoring in our study did not continue into late September, it has been documented by other studies that peaks do occur during that time (*Haven & Fritz, 1985*).

## Environmental effects

Temperature and salinity are known to have an influence on oysters throughout their life cycle (*Hori, 1933*; *La Peyre et al., 2013*). Although there was no correlation found between larval counts and temperatures, settlement was greatest between 22 and 26 °C at DNR Pier, Guys Point, Queen Sound, and Wachapreague (data not shown). These temperatures were within the optimal range (20 and 32.5 °C) for oyster larval growth (*Calabrese & Davis, 1970*) and the ambient water temperatures (20 and 30 °C) that induce adult oysters to spawn (*Horn Point Oyster Hatchery, 2021*). Previous studies observed increased settling with thermal shock (*Lutz, Hidu & Drobeck, 1970*; *Hidu & Haskin, 1971*). Perhaps the mixing and change in temperature between the warmer bay water and cooler ocean water from the inlets may have contributed to settlement.

The optimal salinity ranges of 12–28 ppt (*Dame, 1996*) and 15–20 ppt (*Barnes et al., 2007*) have been reported for oyster growth. Our results are consistent with *Nelson (1923)*, who observed the greatest abundance of straight-hinge larvae at stations in the most saline and lower portion of Barnegat Bay, New Jersey. This is contrary to laboratory experiments by *Hidu & Haskin (1971)*, in which oyster larvae were not stimulated to settle with an increase in salinity. In our study, the highest larval settlement occurred at sites with average salinities >30 ppt over the 2019–2020 period, which is just above the optimal range for oyster growth. Salinity had the strongest and most significant correlation with larval counts. Although these sites were also located closest to the inlets, where salinity is naturally higher, there are other factors such as current flow and/or tidal excursion that can influence settlement.

The negative correlations between larval counts, pH, and DO were significant, but the correlation coefficient was weak (<0.20). For oysters, the ideal range in pH and DO for growth is 6.75 to 8.75 (*Calabrese & Davis, 1966*; *Clark & Gobler, 2016*) and 7 mg O $L^{-1}$, respectively (*Clark & Gobler, 2016*). The average pH at DNR pier, Mills Island, Guys Point, Queen Sound, and Wachapreague fell within the desired range for oyster larvae, but not for DO. Interestingly, the site with the lowest average DO was at Wachapreague (4.6 mg O $L^{-1}$), which had the second greatest larval counts. This further corroborates the negative correlation between larval counts.

## CONCLUSIONS

This is the first recruitment study for oyster larvae in the MCBs, and the resulting spatial and temporal distribution patterns can provide insight into evaluating restoration initiatives and serve as a foundation for future recruitment studies in other lagoonal estuaries. This study resulted in four significant findings: (1) ceramic tiles received significantly greater recruitment than PVC plates, (2) new recruits settled in the greatest numbers at sites that were closest to Ocean City and Chincoteague inlets, as opposed to sites further within the bays, (3) settlement occurred between late June and early July into mid-August, which was consistent with previous studies at similar latitudes (*Shaw, 1967*; *Kennedy, 1980*), and (4) the spatial and temporal patterns of settlement were essentially identical in both 2019 and 2020, although recruitment was four to five times greater in 2020. These results can supplement ongoing data collection (*e.g.*, surveys of fish, shellfish, submerged aquatic vegetation, water quality, and current drift monitoring) to gain a broader understanding of the MCBs and provide baseline data upon which to build. Notably, it may guide stakeholders in evaluating the decision to potentially pursue an oyster restoration project within the MCBs and similar lagoonal estuaries.

### Recommendations

For any oyster restoration project, monitoring is recommended prior, during, and after restoration to assess the reef habitat, the organisms living on the reef, and interactions among organisms (*Thayer et al., 2005*). This is important so adjustments can be made as needed and the progress of the restoration can be observed over time. Examples of restoration techniques used in the Mid-Atlantic and other areas include creating 3-dimensional structures with vertical relief that emulate natural oyster reefs. These consist of clutch, which is a material (*e.g.*, shells, shell fragments, limestone, concrete, *etc.*) used to build attachment substrates for oyster larvae (*Kurz, 2012*). Clutch is deposited on the sediment or a foundation then are piled to make the vertical structure (*Luckenbach, Mann & Wesson, 1999*; *O'Beirn et al., 2000*; *Brumbaugh & Coen, 2009*). Other methods include pre-cast limestone or concrete structures (*e.g.*, oyster castles or oyster balls) and oyster shells in bags (*Olander et al., 2020*; *Virginia Institute of Marine Science, 2023*). Partnering with local agencies as well as oyster farmers and watermen would aid in the collection of necessary data.

For setting up a larval recruitment and settlement monitoring study, ceramic arrays would be the best sampler to use in the short term (a spawning season), prior to restoration, to evaluate the location and time to establish a restoration project. The ceramic arrays should be utilized during at least two spawning seasons to determine if there are spatial and temporal patterns in settlement. Oyster shells should be prioritized for use in restoration efforts over ceramic substrate because of the protein periostracum present on oyster shells (*Crisp, 1967*), chemical cues released from conspecifics (*Tamburri, Zimmer-Faust & Tamplin, 1992*), and the contoured surface (*Taylor, Southgate & Rose, 1998*), but ceramic tile is an alternative if oyster shells are not readily accessible. Natural recruitment of wild oyster larvae aid in restoration success by supplementing restoration efforts (*Schulte & Burke, 2014*). *Schulte & Burke (2014)* concluded that restored reefs with

planted oyster larvae and adults recruited populations with greater densities than unenhanced reefs because wild settlement increased populations by two to three orders of magnitude. The additional larvae and adults provide additional substrate for wild oysters in which to settle (*Southworth & Harding, 2014*).

Based on the results of our study, we recommend the following for site selection prior to pursuing oyster restoration in the MCBs and other lagoonal estuaries: (1) selecting a restoration site that is in close proximity to broodstock, has a slow flushing rate, and circulation patterns that retain larvae, (2) utilizing oyster shells as substrate for preliminary recruitment studies and/or restoration projects (if oyster shells are not accessible, a ceramic array design as seen in this study, can be used as an alternative), (3) establishing a restoration site prior to or in early June (in the Mid-Atlantic) to ensure wild oyster larvae settle during peak time, (4) conducting additional research on the current state of parasites, overwintering, and diseases to ensure survival and growth of oysters, and (5) establishing a monitoring program to assess progress and address environmental changes (recommendations further described in *Kennedy et al., 2011*).

## ACKNOWLEDGEMENTS

This article is in honor of Daniel Cullen, who was an incredible friend, mentor, and colleague. He is missed greatly.

We would also like to thank our boat Captains for their assistance in transportation to our study sites as well as the National Park Service, Maryland Department of Natural Resources, and Maryland Coastal Bays Program in guiding the location of study sites. We are grateful to Elizabeth North, Mitchell Tarnowski, and the anonymous reviewers for providing comments to improve this manuscript. Special thanks to Steve Doctor and P.G. Ross for sharing their expertise as well as peers at the University of Maryland Eastern Shore for assisting in the field and laboratory. Lastly, we thank Cara Schweitzer at the NOAA NMFA-Office of Habitat Conservation for her feedback on the statistic models.

### Funding

This project was supported by the National Science Foundation (NSF) Center of Research Excellence in Science and Technology (CREST) award (#1547821) and by an award from the National Park Service (Task Agreement #P18AC01303). The funders had no role in study design, data collection and analysis, decision to publish, or preparation of the manuscript.

### Grant Disclosures

The following grant information was disclosed by the authors:
National Science Foundation (NSF).
Center of Research Excellence in Science and Technology (CREST): #1547821.
National Park Service: #P18AC01303.

## Competing Interests

The authors declare that they have no competing interests.

## Author Contributions

- Madeline A. Farmer conceived and designed the experiments, performed the experiments, analyzed the data, prepared figures and/or tables, authored or reviewed drafts of the article, and approved the final draft.
- Sabrina A. Klick analyzed the data, prepared figures and/or tables, authored or reviewed drafts of the article, and approved the final draft.
- Daniel W. Cullen analyzed the data, prepared figures and/or tables, authored or reviewed drafts of the article, and approved the final draft.
- Bradley G. Stevens conceived and designed the experiments, authored or reviewed drafts of the article, and approved the final draft.

## Field Study Permissions

The following information was supplied relating to field study approvals (*i.e.*, approving body and any reference numbers):

Field experiments were approved by the Maryland Department of Natural Resources under Scientific Collection Permit numbers SCP201964 and SCP202091.

## Data Availability

The data is available at SEANOE: Farmer Madeline, Cullen Daniel, Stevens Bradley (2022). Eastern Oysters *Crassostrea virginica* settle near inlets in a lagoonal estuary: Spatial and temporal distribution of recruitment in Mid-Atlantic Coastal Bays (Maryland, USA). SEANOE. https://doi.org/10.17882/86685.

## Supplemental Information

Supplemental information for this article can be found online at http://dx.doi.org/10.7717/peerj.15114#supplemental-information.

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
