# Peer review of "Eastern oysters Crassostrea virginica settle near inlets in a lagoonal estuary: spatial and temporal distribution of recruitment in Mid-Atlantic Coastal Bays (Maryland, USA)"

_PeerJ, doi:10.7717/peerj.15114_

## Round 0.1 · original submission · Major Revisions

Dear Dr. Farmer and co-authors,

Thank you for your submission to PeerJ.

After reading carefully the comments by the three referees, I reached the decision of major revision. Please, refer to the reviewer’s comments and answer and modify the manuscript accordingly. Moreover, I personally made some comments to the manuscript. I found it extremely interesting, but several points I think should be improved, not least the statistical one. I attach my comments below.

Lines 169-170: In line 164 you wrote that dimensions were different. Can you please be clearer when giving plates and tiles dimensions?
Lines 185-192: It is not clear if this part is an analysis comprised in the present MS or it makes part of a preliminary test you made before the tests for it.
Lines 187-189: It is not clear what was the effect of the cage around the PVC tiles. Could it have had some effect on settlement? I suspect that a collector design could be a big bias. I didn’t see any collector effect analyzed in your statistical model neither a counterpart with ceramic plates.
Line 191: If your tests already found a better effect from the ceramic tile, why did you assess again PVC?
Lines 203-205: So, not all sites were covered by the experiment. There is a quite big unbalancing among sites sampled with collectors and arrays.
Line 212-214: I suspect here a possible pseudoreplication problem. In order, your sampling unit should be the group of the three tiles, as they are strictly linked each other and influenced each other. I give a possible solution some rows below.
Line 219: Was Secchi’s disk appropriate for such a shallow depth?
Line 223: It is not clear, how did you transport them to the laboratory?
Line 227: Why only underside for ceramic and on both sides for PVC?
Lines 237-239: So, it is not clear, why considering also 2019 in material and methods? And, why all the discussion on the A side and the B side? Probably, several information is useless is not well motivated.
Line 240: See comment for line 170.
Line 249: “Although this did not completely normalize…” Did you try to assess normality of residuals after model application? Sometimes transformation is not necessary as other statistical techniques (if needed) are available.
Lines 252-255: Why two models and not one model only including both info? Another question is: why didn’t you use environmental variables also for the models? Moreover, following the comment at lines 212-214 it should be better adopting a mixed model with the single tile as random effect to adjust for the pseudoreplication issue. Moreover, see comment to figure 5 as a general comment for the transformation problems leading to statistical assumption violations. Anything is reported about the diagnostic of the model. Did you apply any test for it? Visual consideration of residuals scattering vs fitted values or distribution? The sentence at lines 254-255 has been already said. Please delete it.
Line 256: After all said, are you sure that Gaussian distribution is the better choice?
Line 259: But results for both sites showed also 2019.
Line 264: I don’t understand how did you apply the GLM here? You didn’t show any results from it.
Lines 266-267: At lines 185-192 you reported the same conclusion as here. So, again my question is: those results make part of the present MS analyses? Fig.4, I don’t know if the boxplot is the best choice to represent these data. Maybe a bar plot with SE bars would give a better sight. Moreover, with stacked bars you will have only one plot for all sampler together.
Lines 270-273: I wonder if it makes any sense to consider this “doubled side” analysis for the PVC, that has been already assessed as less effective in oyster settling.
Line 278: “…in comparison to the remaining sites.” A table with all results should be reported.
Line 282: And this is the reason why you should consider both space and time in the same model, because a possible interaction can occur between them.
Line 351: But only 6 are evaluated.
Figure 5: It seems that lots of data are fall into the outlier points. Moreover, all these outliers or extreme values seem to me as a violation of variance difference assumption due to skewness that even the log transformation didn’t solve.
Figure 6: A high variability exists, and this could be due to the fact that you are considering tiles or plates as simple unit. Moreover, by pooling all months you do not consider the seasonality or some difference in reproduction and recruitment. Another reason why the statistical analyses should be improved and refined.

Reviewer 1 ·

Basic reporting

The authors present an interesting manuscript about spat settlement around the Maryland coastal bays. The research has value and I appreciate the challenges in both the spatial and temporal scope and then the pandemic. The manuscript is generally well written, although there are times in the introduction became repetitive, certain areas of the text that are probably unnecessary or at least could be trimmed, all the figures may not be necessary, etc. Otherwise, this manuscript follows the basic reporting guidelines. Below are some examples of areas that can be addressed in revisions:

LN 87-91 – this paragraph is only 2 sentences. Consider filling out the paragraph or combining with another paragraph in the introduction
LN 106-112 – this is almost a word-for-word repetition of LN 68—74, so does not need to be repeated here
LN 158-162 – While I appreciate the explanation, I don’t think this text is necessary
Results - I had a really hard time with the Spatial Distribution portion of the results. I think this section can be shortened and clarified, perhaps by entering data into a table that has all sites and both years
Discussion - Like the results, I think the Spatial Distribution portion of the discussion could be trimmed. For example, the flushing time and circulation paragraphs could probably be trimmed and combined. More specifics from the discussion below:
LN 393-398 - seems out of place in this paragraph which is referencing the top vs bottom patterns
LN LN 401-409 - I do agree that there are broad-scale patterns in recruitment of organisms like oysters, but there is a lot of literature that suggests that within a given location, where you place collectors could influence recruitment to those collectors (i.e. the landscape/seascape setting). That could confound the broad spatial patterns. Might be something to think about i.e., were there no recruits or were they just missed.
LN 415-421 - this is an example of 4 sentences that can likely be condensed down into 1 or 2 sentences
LN 453-471 - the two paragraphs can each be condensed and combined
Figures - Figure 4, 5, 6 and 7 all relaying similar and related information and are not all necessary. My suggestion is that Figure 4 can probably be eliminated since Figure 5 shows the different collector types and figures 6 and 7 show the spatial distribution. I also think selecting one of Figure 6 or 7 is also probably sufficient, rather than showing both.

Experimental design

The research questions are well defined. The experimental design seems fine although I did find it difficult to follow along with everything. Some questions to consider:

1) I am not clear on what is really the difference between the collector and the array other than being inside a wire cage? Since the collector and array both had PVC plates, I am just curious why they are considered different.

2) Why were the plates cut to be different sizes than tiles? I am also a little confused regarding ln 172-173 and the border to define a smaller counting area? I assume this is to eliminate any potential edge effects? But its not clear why the outer border was different on the two sides and also whether the ceramic tiles also were treated the same way.

3) I am not sure what line 200-201 means. If I understand the whole methods correctly, in 2020, you selected just 6 sites that exhibited some recruitment in 2019. Is that correct? What does it mean that the planned five swap dates could not be deployed? Were they left out longer in 2020? Were some sets deployed longer than others? This was unclear.

4) I appreciate that WQ data was collected, but have the authors considered an analysis that incorporates the WQ? Currently, it appears to be only descriptive. That's fine, but it might be useful to help explain some of the spatial variation in recruitment if one or more WQ factors were shown to be correlated to recruitment.

I think addressing some of these issues would help clarify the methodological approach.

Validity of the findings

The authors meet the requirements for this section. As mentioned in previous areas, I think improvements in the text and the methods will help the overall usefulness of these findings. Recruitment is a tricky subject that varies over space and time and it is always hard to know if something really isn't there, or just missed. I think the authors do a good job in describing what their results mean within the context of their questions - even if some parts of the discussion can be reduced. Particularly because this manuscript has a management implication, I do think the authors should consider and try to conduct analysis that incorporates the spatial variation in WQ with recruitment, as that could help inform restoration efforts.

Reviewer 2 ·

Basic reporting

-The authors’ writing style was clear an easy to follow. The manuscript was well-organized and followed a logical flow. The goals were clearly identifiable thanks to the numbered points, which was replicated in the results of the abstract and discussion of the main manuscript. Overall, the presented background info was sufficient and helped focus the broader impact of the manuscript.
-One minor wording change that might be more intuitive is to use the phrase “slow flushing rates” as opposed to “long flushing rates”.
-Another minor wording suggestion would be to refer to plate sides as top/smooth and bottom/rough sides instead of A and B so that readers do not need to memorize which side is which throughout the manuscript.
-Since there are so many sites and variables that are considered throughout the manuscript, extra details added to the map in figure 1 would be greatly improve the readability of the manuscript. For example, would it be possible to add zones to the map where oysters are currently found and where they have been historically found? If the historical zone covers most of the MCB on the map, then perhaps this can also be stated more explicitly in the manuscript (L106-112).
Can the bays listed on L115-119 also be added to the map in Figure 1 since these bays are also later referenced in the discussion section? This would be incredibly helpful for readers that are not familiar with this geographical area. Finally, James River is also mentioned on L410, would it be possible to also add the river and river mouth to the map in fig 1? Or maybe just an arrow pointing to river mouth? This would also enhance understanding of the authors recruitment results.

Experimental design

-The addition of repeating the study over multiple years strengthened this study by accounting for potential differences in recruitment among years. The research aims were well defined throughout the manuscript and a similar structure was repeated in the results and discussion which was incredibly helpful for the reader to follow throughout the manuscript. Since this is also the first study to look at oyster recruitment distributions in MCBs, it is an incredibly important knowledge gap that is filled for future restoration efforts.
-L158-160: the clear definitions for “settlement” and “recruitment” greatly strengthened the manuscript and the repeatability of their methods.
-There could be more detail added to the statistical methods for the generalized linear models used. For example, in paragraph L251-262, please add somewhere what were the fixed and random effects in the model. Were replicated sampler types at each site a random effect? Were tiles/arrays on each sampler nested within replicated sampler types as random effects? Please clarify how these models were run.

Validity of the findings

-The manuscript would be improved with a brief statement or paragraph in the result section describing the main differences in environmental variables among all the field sites. Which sites had the highest and lowest temperatures/salinity/DO/depth? Could the sites be grouped into different categories based on environmental variables or geographical regions (i.e. bays vs inlets). This would help with interpretation of results. The authors also mentioned they measured DO at the sites, but this result is not presented in Table 1. Can these data be added? I would expect DO to be an interesting variable for oyster recruitment with lower recruitment seen in areas with lower DO.
-The broader implications of this manuscript could be further enhanced in the discussion section with a discussion of the utility of the different samplers used for restoration efforts. Is there some connection to how these different recruitment samplers work for long term conservation and restoration? Once oysters have recruited, do they build substrate for subsequent oysters? How long does it take for the samplers to degrade or fall apart? Can they work long term over multiple years to restore an area? In the recommendation section, you mentioned that oyster shells should be prioritized over ceramic tiles. Is there an explanation and citation that can be added here to justify this sentence (L502)?
-L415-419: It seems that oyster recruitment is higher near inlets but also areas of longer flushing rates. This may seem counterintuitive at first, but I think it makes sense, it just needs a little more explanation in discussion to make this result clear.
-L437-439: What about distance to the closest oyster source population as an explanation for higher recruitment near inlets? If there are none within the shallow bays, they must be coming from outside the Bay. Please touch on this potential explanation here.

·

Basic reporting

The scientific reporting is acceptable. The text is clearly written.

Experimental design

The study design is well thought-through and the research generally fills a void in the literature for the Eastern oysters.

The statistical method needs a closer critical look. Whereas the standardisation with a formula is good, it has the likelihood to parameterise the the non-parametric (count) variable [recruitment]. This is very critical to the rest of the analysis. As a non-parametric path was chosen, the authors ought to maintain the recruitment as a non-parametric variable and further maintain a non-parametric measure of central tendency throughout the text. The authors may want to justify their approach.

Validity of the findings

Some revisions have been suggested for the statistical approach. Once that is checked or justified, the findings should be valid. Otherwise, the study is novel to the region and the conclusions are related to the research question.

Additional comments

No additional comments

---

## Round 0.2 · Minor Revisions

Dear authors, thank you very much for the big effort made to improve the manuscript. Please, now refer to the comments made by reviewer 1 and my personal notes listed below. I have a question on the zero inflated zeros and overdispersion (this is something emerging both from the answers to my previous comments and to other referees). Although they represent an issue in ecological data such as the ones reported here, several remediations are available to solve them (or try to). I think, for example, that negative binomial distribution potentially can reduce the overdispersion you faced. Moreover, zero inflated models and techniques are available to reduce the influence of an excess of zeros (I think about ZIP or ZINB models that can solve both problems). I know they are complicated models to apply, but did you try to apply them? I think these techniques can give a more precise picture of the oyster settling and recruitment. Otherwise, the results you obtained should be taken with caution. In case you are not able to apply such models, please write a small sentence on this aspect.

Reviewer 1 ·

Basic reporting

The authors present an improved and useful manuscript regarding settlement and recruitment of eastern oysters in Maryland's coastal bays. The manuscript is well-written and easily readable. It is clear that the authors made major changes based on feedback from the Reviewers which improved the overall manuscript. There are a couple of instances that might benefit from some minor tweaks (see below), but overall, this manuscript meets the basic reporting requirements.

LN 96: either here or in the discussion, it might be helpful to know a little bit about what these restoration initiatives might be and/or how restoration is typically done in the region for readers who are not familiar.

LN 353-354: I am not sure that is the interpretation I would make from the previously listed studies. I can't see how sanding would have counteracted conditioning, if anything, it would have increased the surface area for the microbes to condition the PVC.

LN 366-375: I think the statements at the end of this paragraph are contradictory to the statements at the beginning. That is, either Chincoteague site has high flushing (367-368) or it has higher retention (later in the paragraph), but it cannot be both highly flushed with a long residence time.

LN 401-409: I am not sure this paragraph is necessary, since fouling/competition is not mentioned elsewhere in the MS, and if the authors choose to keep it, I don't think it fits in this section.

Experimental design

The research questions are well-defined, and with the edits, the experimental design is easier to follow along. The authors made clear and concerted efforts during revisions to explain some questions about the approach posed by all reviewers, and also included new analysis.

One thought - as the authors clearly define and differentiate "settler" from "recruit" it might be beneficial for readers to include an side by side image with scale and descriptions of what was being called a larval settler (pre-metamorphosis?) vs and up to 2 week old recruit.

Validity of the findings

As I mentioned in my initial review, I commend the authors for tackling questions about recruitment in an area without any real prior data, specifically because of the highly variable nature of the measurements and the difficulty in analyzing the data. The revisions greatly improved the manuscript, and I think the authors do a better job in the discussion than on the earlier version.

·

Basic reporting

In line with my previous review, I maintain my verdict on the basic reporting - the text is clearly written and the scientific reporting is acceptable. The authors have improved the structure of the reporting.

Experimental design

The study design is well thought-through and the research generally fills a void in the literature for the Eastern oysters.

Validity of the findings

The authors have done an extensive revision of the data analysis. The issues of concern around inconsistency and the mixture of parametric and non-parametric analytical methods for the same response variable type, i.e., recruitment, have been rectified. The non-parametric approach is maintained and adhered to, and the median is reported. The revised approach is acceptable.

The authors, however, report S.E. after the median in some instances - my question is, are these really S.E. of the median? If so, then the description of Figure 5 should be particularly explicit about that. The sentence... "Bars above median counts represent standard error (SE)" in the figure caption should be completed with "of the median". If otherwise, then it should be clarified. This can be treated as a minor edit that can be handled during proofing.

---

## Round 0.3 · accepted · Accept

I congratulate the authors for their extraordinary work. I think that after all the changes suggested, the manuscript greatly improved and in its present version it is ready to be published and interesting for the general audience.